# Injury Incidence Increases after COVID-19 Infection: A Case Study with a Male Professional Football Team

**DOI:** 10.3390/ijerph191610267

**Published:** 2022-08-18

**Authors:** Antonio Maestro, David Varillas-Delgado, Esther Morencos, Jorge Gutiérrez-Hellín, Millán Aguilar-Navarro, Gonzalo Revuelta, Juan Del Coso

**Affiliations:** 1Faculty of Medicine, Oviedo University, 33003 Oviedo, Spain; 2Hospital Begoña, 33204 Gijón, Spain; 3Exercise and Sport Science, Faculty of Health Sciences, Universidad Francisco de Vitoria, 28223 Pozuelo, Spain; 4Clinica El Molinon, Centre for Advanced Medicine, 33203 Gijón, Spain; 5Centre for Sport Studies, Rey Juan Carlos University, 28943 Fuenlabrada, Spain

**Keywords:** SARS-CoV-2, injuries, epidemiology, football (soccer), professional

## Abstract

Background: The SARS-CoV-2 virus disease has caused numerous changes in sports routines in the last two years, showing the influence on an increase in sports injuries. The aim of this study was to prospectively analyze the incidence and characteristics of injuries in male professional football players diagnosed with COVID-19 when they return to play after recovering from this illness. Methods: Injury characteristics of professional male football players were recorded for the 2020–2021 season following the international consensus statement from the International Olympic Committee (IOC). SARS-CoV-2 infection in the football players was certified by PCR analysis. Injury epidemiology was compared in players infected by the SARS-CoV-2 virus before and after being diagnosed with COVID-19. Results: 14 players (53.8%) were diagnosed with COVID-19 during 2020–2021 season and 12 (46.2%) were not infected (controls). Only three (21.4%) had suffered an injury before being diagnosed with COVID-19. Eleven players (78.6%) had injuries after being diagnosed with COVID-19 (*p* < 0.001). Among the players diagnosed with COVID-19, injury incidence increased on their return to play after the infection (3.8 to 12.4 injuries/1000 h of exposure, *p* < 0.001). Additionally, injury incidence during training (10.6 vs. 5.1 injuries/1000 h of exposure, *p* < 0.001) and matches (56.3 vs. 17.6 injuries/1000 h of exposure, *p* < 0.001) was ~two-fold higher on return to play after COVID-19 compared to controls (33.4 vs. 17.6 injuries/1000 h of exposure, respectively, *p* < 0.001). Conclusions: Injury incidence in professional football players who had been infected by the SARS-CoV-2 virus significantly increased compared to the injury rates that these same players had prior to the illness. Additionally, the injury incidence was higher when compared to players who were not infected by the SARS-CoV-2 virus during the season, especially during matches.

## 1. Introduction

Coronavirus disease 2019 (COVID-19), caused by a severe acute respiratory syndrome produced by the SARS-CoV-2 virus, is an illness that can cause multi-system failure and some other life-threatening conditions in a relatively high proportion of infected individuals [1]. In March 2020, COVID-19 became a pandemic, and since then avoiding the spreading of the virus has been a main target of most national and international health authorities. To reduce the spread of SARS-CoV-2 during the first wave of COVID-19, when there was no vaccine against COVID-19 available in the market, governments of many countries declared home-confinement measures in addition to other isolation and hygiene measures such as social distancing and the use of face masks. In the first wave, most professional sporting activities were suspended for the home-isolation period, causing an unprecedented disruption in sport and training routines [2,3,4]. Due to the potential risks of infection, most sports competitions in Europe were not resumed after the first wave of COVID-19 but the enormous economic revenues and popularity of football meant that most national football leagues were resumed to finish the 2019–2020 season with extraordinary measures to avoid infection and injury [5,6]. 

In the subsequent waves of COVID-19, the approach of health and football authorities to protect football players from contagion during training, competition and travelling has been different. First, although the fixture of the national football leagues in the 2020–2021 season was similar to previous seasons, some changes of in-game regulations were allowed, such as the five-substitutions option [7,8]. Additionally, players had to undergo several PCR or antigen tests per week, especially before a match, to aid in the early detection of COVID-19 in the football team and to avoid virus spread, among other measures [5,6]. The competitions were maintained as it was then known that the risk of on-field transmission of SARS-CoV-2 is very low, while the source of infection in football players is most likely not related to activities on the pitch [9]. Despite the strong actions established by health and football authorities to avoid infection during the 2020–2021 season, hundreds of professional footballers from European leagues were diagnosed with COVID-19, including several outbreaks in professional football clubs. Habitually, professional football players do not have severe symptoms after SARS-CoV-2 infection and recover from COVID-19 within 2 to 3 weeks. However, it is unknown how infection can affect performance or injury rate when the players return to play. 

Several investigations have been geared to determining the effect of the COVID-19 pandemic on injury incidence in professional football. Two of them analysed the effect of football suspension and home isolation in the 2019–2020 season in the German Bundesliga [10,11] and they both concluded that athletes did not experience an increased rate of injury when the competition was resumed. However, the scenario of competition suspension and home isolation has not been replicated in the 2020–2021 season and the outcomes of this investigation are not applicable to the current context of professional football. Recently, Orhant and colleagues [12] presented an analysis of epidemiological data from the French leagues 1 and 2, comparing the 2020–2021 season completed during the COVID-19 pandemic and a regular season (2018–2019). They concluded that the COVID-19 pandemic does not influence injury occurrence and patterns in French professional football. However, the analysis used in these three investigations merged injury data from players who had been diagnosed with COVID-19 and players who had not had the illness, and therefore it is impossible to ascertain how COVID-19 affects a professional football player once he/she returns to play after recovering from the illness. To adequately investigate the effect of COVID-19 on injury incidence, a comparison of injury rates before and after SARS-CoV-2 infection must be made and analysed, which was not done in any of the aforementioned studies. Indeed, with better sample size the conclusions of these studies would be much stronger.

Therefore, the aim of this study was to prospectively analyse the incidence and characteristics of injuries in male professional football players diagnosed with COVID-19 when they return to play after recovering from this illness.

## 2. Materials and Methods

### 2.1. Participants

Twenty-six professional football players, from a team competing in the LaLiga Smartbank (second division of the Spanish professional football league), volunteered to participate in this investigation. The following inclusion criteria were established for participants: (a) football players with a first team contract; (b) regular exercise training of >1 h per day, >5 days per week for the prior six months; and (c) had to not be admitted to hospital due to COVID-19 infection. The following exclusion criteria were applied to potential participants: (a) several injuries within the prior two months; (b) injuries incapacitating football training or matches in the previous six months; (c) mechanical instability of the knee joint [13]. An a priori sample size calculation indicated that participants of one professional football team were needed to obtain statistically significant differences in injury incidence rate suffering COVID-19 with respect to non-COVID-19 in all sessions. This a priori sample size was calculated to obtain an effect size of 0.92 in Cohen’s d units (statistical power of 80% with type I error set at 5%), based on a previous investigation that obtained this effect when they study the effect of incidence rate with COVID-19-affected professional football players across 12 countries. [14]. The required sample size was determined using G*Power software (Universität Düsseldorf, Düsseldorf, Germany) [15]. From the sample, three were goalkeepers, eight were defenders, eight were midfielders and seven were attackers. The players trained an average of 8.9 ± 0.3 h/week during the season and performed ~1 competitive match per week, for a total of 40 official matches during the whole 2020–2021 season. As a result, total exposure time was 11,320 h (10,655 h of training and 665 h of matches) (Table 1). Before taking part in the study, the players were fully informed about the risks and benefits of participating in the investigation and provided their written informed consent.

The experimental procedure of this study was in accordance with the latest version of the Declaration of Helsinki and was approved by the Francisco de Vitoria University Research Ethics Committee (32/2020) and complied with the Declaration of Helsinki for Human Research of 1964 (last modified in 2013).

### 2.2. Confirmation of SARS-CoV-2 Infection in the Football Team

Following the guidelines provided by Spanish football and medical authorities, all players were tested for SARS-CoV-2 infection using a rapid antigen test four times per week and by Polymerase Chain Reaction (PCR) analysis 48 h before any official match for the duration of the season. In those cases of a positive rapid antigen test, infection by SARS-CoV-2 was confirmed by a PCR analysis. Once a positive case of COVID-19 was reported in a football player, the medical staff of the team recorded the date of diagnosis, the symptoms suffered by the player during the infection, the quarantine time set and the duration of the illness, including the dates of return to training and competition. The medical staff considered that a player had overcome the infection once they had had two negative PCR tests over a 24-h period.

No player was vaccinated when infected with COVID-19 in the 2020–2021 season.

### 2.3. Injury Data Collection

Injury data were obtained prospectively during the 2020–2021 competitive season. All injuries were diagnosed, classified and recorded by the medical staff of the football team using the classification system developed by the medical commission of the International Olympic Committee (IOC) [16]. The medical staff of the club recorded injuries during the season. In each recordable injury, the region, body area, type of injury, tissue affected, and pathology type were recorded. Injury diagnoses were aided by different clinical methods (e.g., X-ray, ultrasound, X-rays, magnetic resonance imaging, etc.) depending on the type of injury. The mode of onset of injuries was distinguished between sudden and repetitive/gradual onset injuries. Within sudden injuries, the mechanism of onset between contact and non-contact injuries was also recorded. Training and competition injuries were also catalogued according to the IOC consensus statement, and its definition was used to classify injury severity. Afterwards, injuries were grouped as minimal (1–3 days), mild (4–7 days), moderate (8–28 days), or severe (>28 days) [17]. Injury prevalence was calculated as the number of players who were diagnosed with an injury during the season. Injury incidence was individually calculated for each football player, including an overall calculation of injury incidence with all injuries divided by the total time of exposure during the season and matches, and training injury incidence by using data of match and training injuries and exposure times in the total squad [18].

### 2.4. Statistical Analysis

All statistical procedures were conducted using SPSS (version 25.0; IBM Corp., Armonk, NY, USA). The normality of each variable was initially tested with the Shapiro–Wilk test, and, consequently, parametric statistics tests were used to determine differences among groups. Baseline characteristics, measured as continuous variables, were calculated as mean values with corresponding standard deviations. Ordinal or categorical variables, such as number of players with an injury or injury severity, were presented as absolute numbers and percentages. In those players with a record of COVID-19 infection, all injury variables (i.e., prevalence, incidence, and injury characteristics) were compared before (PreCOVID-19) and after infection (PostCOVID-19) for a within-subject comparison using a *t*-test for paired samples and χ^2^ test. Additionally, injury variables during the season were compared among players with a record of COVID-19 infection and the players who did not report COVID-19 infection during the season (between-subject comparison), who acted as controls, using a *t*-test for independent samples and χ^2^ test. For each variable, 95% confidence intervals (95% CI) were obtained using the Poisson model. The significance level was set at *p* < 0.05 for all statistical calculations.

## 3. Results

### 3.1. Infection and Symptoms of COVID-19

During the season, 14 players (53.8% of the total sample) were diagnosed with COVID-19 infection while the remaining 12 players never tested positive in any antigen or PCR test. From the sample of players with a recorded COVID-19 infection, five players (35.7%) did not report any symptoms while the remaining nine players reported one or several symptoms such as fatigue, asthenia, fever and headache (Table 2). No player had to be admitted to hospital due to COVID-19 infection and average time of quarantine until PCR testing was negative for SARS-CoV-2 was 16.6 ± 4.9 days, performing physical maintenance scheduled by the team’s physical trainer. Return to train after COVID-19 infection was 17.9 ± 4.4 days and return to play was 20.1 ± 3.1 days. Figure 1 contains information of the time of infection and evolution of COVID-19 for each of the players who tested positive during the 2020–2021 season.

### 3.2. Injury Circumstances

One of sixteen muscle and tendons injuries were traumatic with an acute onset. Overuse injuries with a gradual onset were more common among knee injuries than among injuries to the lower back, foot, and hand muscles (50% vs. 16.7%, 16.7%, and 16.7%, respectively, *p* < 0.001). There was only one nerve injury due to L4-L5 disc protrusion. Almost all muscle injuries occurred in noncontact situations (adductors18.2%; quadriceps and hamstrings, 72.7%; and calf muscles, 9.1%), and only 37.5% occurred during matches.

### 3.3. Injury Incidence in Players Diagnosed with COVID-19 in Comparison to Controls

From the 14 players diagnosed with COVID-19, only three (21.4%) had suffered a football-specific injury before being diagnosed, while 11 players (78.6%) had injuries after being diagnosed with COVID-19 (*p* < 0.001) during the 2020–2021 season. The injury incidence for the whole season was similar in players who had had COVID-19 and in controls (*p* = 0.527). For the whole season, training injury incidence was also similar in players with COVID-19 and in controls (*p* = 0.276), but the incidence of match injuries was ~two-fold higher in players who had COVID-19 than in controls (*p* < 0.001), not showing differences about inactivity and return to play (*p* = 0.632). The analysis of the subsample of players who had been infected with COVID-19 indicates that, when comparing injury rates before and after infection, the overall injury incidence and training and match injury incidence increased after COVID-19 infection when compared to the period prior to infection (all *p* < 0.001). Lastly, injury incidence after COVID-19 was higher than that reported by the control group for the whole season (all *p* < 0.001; Table 3). 

The same pattern of injury incidences was present when analyzing only muscle/tendon injuries. Muscle/tendon injury incidence for the whole season was similar for the players diagnosed with COVID-19 and for the controls (*p* = 0.416, Table 4). For the whole season, muscle/tendon injury incidence during training was also similar for the players who had suffered COVID-19 and for the controls (*p* = 0.447) but the incidence of muscle/tendon injuries during matches was ~two-fold higher in players who had had COVID-19 than the controls (*p* < 0.001). The analysis of the subsample of players who had had COVID-19 indicates that incidence of muscle/tendon injury overall and during training or competition increased after COVID-19 when compared to the period prior to infection (all *p* < 0.001) or when compared to the control group (all *p* < 0.001).

For the whole team, most of the injuries were muscle/tendon injuries, with a lower percentage of joint/ligament injuries (Table 5). The same distribution of injury types was present in players who were diagnosed with COVID-19 and controls (*p* = 0.723). Additionally, the distribution of injury per type was similar in players diagnosed with COVID-19 before the infection and after their return to play (*p* = 0.621). Regarding severity, players diagnosed with COVID-19 needed a median of 22.0 days to return to play (interquartile range = 10.0–25.5 days) before infection and the return to play after the injury was similar after infection 29.0 days, (interquartile range = 16.0–33.5 days). In fact, the return to play after an injury in the players diagnosed with COVID-19 was similar to the controls who needed 21.0 days to return to play (interquartile range = 10.0–28.0 days). No player ended his career due to injury.

## 4. Discussion

In the 2020–2021 season, football leagues were organised with strict measures to minimise the likelihood of contagion of SARS-CoV-2 during training and competition [19]. This meant that most football competitions were held on the scheduled dates with full unrestricted activity associated with COVID-19 [5,6]. However, despite all the measures set for clubs, hundreds of players were diagnosed with COVID-19 as the virus was still present in our society and vaccination did not fully prevent virus transmission. The aim of this study was to prospectively analyse the incidence and characteristics of injuries in male professional football players diagnosed with COVID-19 when they return to play after recovering from this illness. The main finding of this study was that, when comparing injury rates before and after SARS-CoV-2 infection, injury incidence was two-fold higher after COVID-19 than before, indicating that this infectious illness may increase the likelihood of injury in professional football players. The higher rate of injury after COVID-19 was present in both training and competition scenarios and it was primarily associated with muscle/tendon injury as this type of injury was the most prevalent before and after COVID-19. These data suggest that preventing COVID-19 infection in professional football players may be an effective measure for avoiding abnormal injury incidence rates during the season. This is important in the context of professional football, where players rarely had severe COVID-19 symptoms, meaning that the illnesses can be sometimes considered as a mere context where the player must be confined at home for several days. The current data indicate that COVID-19 may influence the return to play of professional football players as they may have a several-fold greater likelihood of injury, mainly a muscle-type injury, during a match.

In the scenario of the 2020–2021 season, players infected with SARS-CoV-2 were rapidly quarantined in their homes while the football teams continued with their normal training and competition routines but with constant PCR and antigen testing to detect more potential cases within the team to prevent a possible outbreak early. While the player was at home, the medical staff of the team contacted him daily to certify the evolution of the illness while the player maintained an in-home training program tailored to the conditions of his place of residence and the evolution of the illness. Once it was certified that the player had no symptoms associated with COVID-19 by two negative results in PCR testing, the medical staff cleared him as free of COVID-19 and the return to training was determined. The strength and conditioning staff prepared a special “return to play” training program to aid the player to recover his physical condition. Again, the training program set for return to play was tailored to the characteristics of the time-out of each player, as the symptoms (Table 2) and duration of COVID-19 (Figure 1) were not equal for all players. In fact, the training program set to aid in the return to play was similar to the ones employed after injuries of moderate severity. Despite these careful measures, the current investigation indicates that players are more prone to injury after COVID-19 than before, although the reasons are not evidenced from this investigation. Interestingly, 35.7% of the players with COVID-19 were asymptomatic while the remaining players had minor symptoms such as fatigue, asthenia, and fever. Therefore, injury probability in the return to play after COVID-19 can be higher even in players with COVID-19 with mild symptoms. 

The outcomes of the current investigation are different to the ones found in previous studies that analysed the effect of COVID-19 on injury rates in professional football [10,11,12]. Seshadri et al. [10] and Krutsch et al. [11] analysed the effect of football suspension and home isolation in the 2019–2020 season in the German Bundesliga, during the first wave of COVID-19. Both studies indicated that injury incidence was minimally affected by the suspension of the competition, with a potential effect of lockdown on match injury rate. However, these investigations cannot be considered to understand the effect of COVID-19 on players’ injury incidence as no case of COVID-19 was detected in the resumption of the German Bundesliga in the 2019–2020 season. Orhant et al. [12] presented an analysis of epidemiological data in the French leagues 1 and 2, comparing a season completed during the COVID-19 pandemic (2020–2021) to a previous season before its onset (2018–2019). Again, their conclusion regarding the absence of effect of the COVID-19 pandemic on injury rate in professional football might be misleading as the analysis merged data on players who had been diagnosed with COVID-19 and players who had not had the illness, and the analysis was for the whole season. In the current study, the whole-season injury incidence of players diagnosed with COVID-19 is similar to controls (Table 3), as this analysis included the rate before the infection. Only when the analysis includes a pre-to-post-infection comparison can we see the true effect of COVID-19 on injury rate. Therefore, although the COVID-19 pandemic may have produced a small and insignificant effect on injury rate in the leagues, mostly because a relatively small portion of players were infected during the season, those players diagnosed with COVID-19 may have a higher likelihood of injury in their return to play after the illness. The outcomes of the current investigation also indicate that the return to training/play of players who have been diagnosed with COVID-19 should be more cautious than routine protocols used for periods of inactivity to avoid a premature return to training/play that leads to injury. It is necessary to establish specific plans to enhance the return to training/play of players who have been infected with SARS-CoV-2. For this, future studies should be carried out to establish the best evidence-based practices in the return to play process after COVID-19 [20]. Beyond injury incidence, the infection of the SARS-CoV-2 virus did not affect other characteristics of the injury such as type, location or severity.

Recently, increased injury rates have been documented following COVID-19 sports suspension in American Major League Baseball [21,22] and the National Football League [23,24]. European reports did not acknowledge the potential complicating factor of the effects of increases in the acute-to-chronic workload ratio (ACWR). Periods of inactivity followed by higher demand activity increases the ACWR and the subsequent risk of injury, founded in several sports [25], including football [26,27], according to data presented in this pilot study with professional football players.

One of the most important findings of this research is that data from professional football players were collected and analysed during the COVID-19 pandemic despite the difficulty in taking these samples in professional athletes in all areas of sport due to their bubble of protection against SARS-CoV-2 infection [28]. The authors followed established epidemiological guidelines for injury comparisons [13,16]. In addition, documentation of the injury was provided by medical experts (physical therapist or medical staff). In the Spanish League, physiotherapists, or medical staff (or both) usually attend all training sessions and are present during matches. The availability of medical staff has improved the accuracy of incidence rates and the correctness of diagnoses.

Despite the strength and novelty of this analysis, the study has some limitations. First, we have included the analysis of only 26 professional football players from a professional football team in the second division of the Spanish National League and from this sample, only 14 had COVID-19 during the 2020–2021 season. However, the data collected in this research were recorded prospectively and controlled by the club’s medical services, assuring the accuracy of injury epidemiological data and exposure times during both training and competition [16]. Further investigations that use this same pre-to-post infection approach should be performed with larger sample sizes. Second, the current investigation was carried out with male football players and lacks analysis of female teams, youth teams, players from other continents, and players playing in different weather conditions and on different types of pitches, which are more of the critical limitations of the study. The analysis of the impact of COVID-19 on injury incidence in female football players is another key element to be studied to understand the true effect of COVID-19 on professional football. Third, only epidemiological data were analysed, not considering other factors that could trigger the apparition of injury, such as perceived fatigue, wellness [29,30] or some genetic variants [31,32]. Despite these limitations, we believe that the investigation contributes to the current knowledge of the effect of COVID-19 on the incidence and characteristics of injuries in male professional football. 

## 5. Conclusions

In summary, injury incidence in male professional football players who had been infected by the SARS-CoV-2 virus, and therefore had COVID-19, significantly increased with respect to the injury rates that these same players had prior to the illness. Additionally, the injury incidence was higher when compared to players who were not infected by the SARS-CoV-2 virus during the season, especially during matches.

## Figures and Tables

**Figure 1 ijerph-19-10267-f001:**
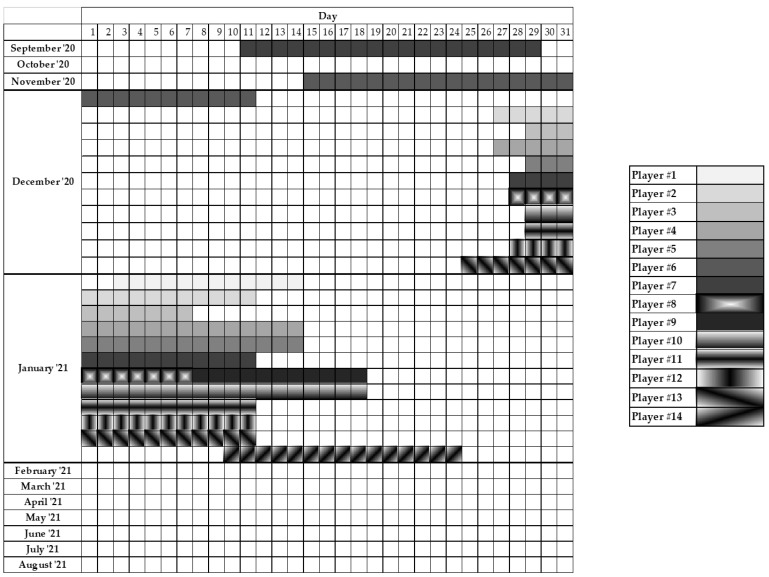
Duration and evolution of COVID-19 infection in professional football players during the 2020–2021 season.

**Table 1 ijerph-19-10267-t001:** General characteristics of the football players at the beginning of the 2020–2021 season.

	Whole Team *n* = 26	COVID-19 *n* = 14	Controls *n* = 12	*p* Value
Age (years)	25.2 (4.9)	23.4 (3.0)	27.3 (6.1)	0.059
Body mass (kg)	76.9 (6.6)	76.2 (5.5)	77.7 (7.7)	0.606
Height (cm)	180.6 (5.5)	180.9 (6.3)	180.3 (4.5)	0.785
Body mass index (kg/m^2^)	23.6 (1.5)	23.3 (1.1)	24.0 (1.8)	0.265
Training exposure (h)	410	421	397	0.169
Match exposure (h)	26	24	29	0.426
Total exposure (h)	438	445	426	0.313

Data are presented as mean (standard deviation) for the whole football team and for players that were diagnosed with COVID-19 in the 2020–2021 season (COVID-19) and players who were not diagnosed with COVID-19 during the entire season (Control).

**Table 2 ijerph-19-10267-t002:** Symptomatology associated with COVID-19 in professional football players with a certified infection of SARS-CoV-2.

COVID-19 Symptomatology	*n* (%)
None	5 (35.7)
Fatigue	6 (42.9)
Asthenia	5 (35.7)
Fever	4 (28.6)
Headache	4 (28.6)
Joint and muscle pain	3 (21.4)
Anosmia	2 (14.3)
Gastralgia, diarrhoea	1 (7.1)

**Table 3 ijerph-19-10267-t003:** Injury incidence during training and competition in players diagnosed with COVID-19 and players who did not have COVID-19 during the 2020–2021 season.

	Whole Season				Pre- vs. Post-COVID-19				
	Whole Team	COVID-19	Controls	*p* Value	Pre	Post	*p* Value	*p* Value Pre vs. Controls	*p* Value Post vs. Controls
Total injury incidence (injury/1000 h of exposure)	6.4	7	5.8	0.527	3.8	12.4	<0.001	0.251	<0.001
	(3.7–9.0)	(4.5–9.8)	(2.4–8.2)		(1.6–6.3)	(4.6–19.6)			
Training injury incidence (injury/1000 h of exposure)	6.2	6.8	5.1	0.276	4.3	10.6	0.001	0.521	<0.001
	(1.6–6.5)	(3.5–8.1)	(2.3–9.1)		(2.7–10.7)	(4.3–16.2)			
Match injury incidence (injury/1000 h of exposure)	26.2	33.4	17.6	<0.001	0	56.3	-	-	<0.001
	(10.5–61.5)	(15.7–57.2)	(9.4–32.6)		(0.0–0.0)	(27.7–89.2)			

Data are incidence (95% confidence interval) for the whole football team and for players that were diagnosed with COVID-19 in the 2020–2021 season (COVID-19) and players who were not diagnosed with COVID-19 during the entire season (Control). In the players diagnosed with COVID-19, the variables are presented for the time period before the diagnoses and after returning to play.

**Table 4 ijerph-19-10267-t004:** Muscle/tendon injury incidence during training and competition in players diagnosed with COVID-19 and players who did not have COVID-19 during the 2020–2021 season.

	Whole Season				Pre- vs. Post-COVID-19				
	Whole Team	COVID-19	Controls	*p* Value	Pre	Post	*p* Value	*p* Value Pre vs. Controls	*p* Value Post vs. Controls
Total injury incidence (injury/1000 h of exposure)	6.8	7.5	5.5	0.416	5.3	11.5	<0.001	0.832	<0.001
	(4.0–9.4)	(5.1–10.8)	(2.52–9.2)		(2.5–9.4)	(6.3–17.5)			
Training injury incidence (injury/1000 h of exposure)	7.1	7.9	6.1	0.447	5.5	12.2	<0.001	0.683	<0.001
	(2.9–10.3)	(4.0–9.7)	(3.3–10.5)		(2.4–10.2)	(7.0–19.3)			
Match injury incidence (injury/1000 h of exposure)	53.2	76.2	36.6	< 0.001	0	81.6	-	-	<0.001
	(35.0–93.7)	(41.6–95.1)	(17.4–52.8)		(0.0–0.0)	(51.3–131.4)			

Data are incidence (95% confidence interval) for the whole football team and for players that were diagnosed with COVID-19 in the 2020–2021 season (COVID-19) and players who were not diagnosed with COVID-19 during the entire season (Control). In the players diagnosed with COVID-19, the variables are presented for the time period before the diagnoses and after returning to play.

**Table 5 ijerph-19-10267-t005:** Injury type in players diagnosed with COVID-19 and players who did not have COVID-19 during the 2020–2021 season.

	Whole Season				Pre- vs. Post-COVID-19		
Injury Type	Whole Team (%)	COVID-19 (%)	Controls (%)	*p* Value	Pre (%)	Post (%)	*p* Value
Fracture and bone stress	0	0	0	0.723	0	0	0.621
Joint and ligaments	7 (29.2)	4 (28.6)	3 (30.0)		1 (33.3)	3 (27.3)	
Muscle and tendons	16 (66.7)	9 (64.3)	7 (70.0)		2 (66.7)	7 (63.6)	
Contusions/skin lesions and lacerations	0	0	0		0	0	
Nervous system	1 (4.2)	1 (7.1)	0		0	1 (9.1)	
Other injuries	0	0	0		0	0	

## Data Availability

Not applicable.

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
