# Peer review of "Injury Incidence Increases after COVID-19 Infection: A Case Study with a Male Professional Football Team"

_ijerph, 2022, doi:10.3390/ijerph191610267_

Round 1
Reviewer 1 Report
The sample size is relatively small. Need to check the normality of the data and if the data is not normally distributed, non-parametric statistical methods should be used in this study.
Author Response
Response to Reviewer comments
ijerph-1857340
Injury incidence increases after COVID-19 infection: a case study with a professional football team
We sincerely thank the Reviewer for carefully proofreading the manuscript and for their helpful and constructive comments. We have addressed all the points raised by the reviewer and have highlighted any change to the manuscript in red using the track changes tool. We believe that our manuscript has been improved by the suggested changes.
The sample size is relatively small. Need to check the normality of the data and if the data is not normally distributed, non-parametric statistical methods should be used in this study.
We welcome reviewer suggestion. In this case, the authors have taken care to confirm the normality of the distributions despite the small sample size.
To ensure this point, the authors have added in the Statistical analysis subsection the following sentence: "The normality of each variable was initially tested with the Shapiro-Wilk test, and, consequently, parametric statistics tests were used to determine differences among groups". Lines 159-161.
On behalf of all co-authors, many thanks for this useful review.

Reviewer 2 Report
1. Inclusion and exclusion criteria are not mentioned in the participant section of Materials and Methods, and the first paragraph of this section should be the content of the results section.
2. Why not study the causes of sports injuries in professional football players?
3. The title should state that it is a study of male professional football players.
4. How was the sample size determined? Is the subject's representative of the target population?
5. The format of Tables 1 through 5 is not three-line, and the horizontal and vertical headings need to be modified.
6. The conclusions indicate that the incidence of injury among professional football players infected with COVID-19 is significantly increased compared to the rate of injury in these players before infection, but there are too many confounding factors, and the evidence is not convincing.
Author Response
Response to Reviewer comments
ijerph-1857340
Injury incidence increases after COVID-19 infection: a case study with a professional football team
We sincerely thank the Reviewer for carefully proofreading the manuscript and for their helpful and constructive comments. We have addressed all the points raised by the reviewer and have highlighted any change to the manuscript in red using the track changes tool. We believe that our manuscript has been improved by the suggested changes.
- Inclusion and exclusion criteria are not mentioned in the participant section of Materials and Methods, and the first paragraph of this section should be the content of the results section.
Thank you for the reviewer's suggestion. The authors have added inclusion and exclusion criteria to provide more information on the sample and to avoid possible selection biases. "The following inclusion criteria were established for participants: (a football players with a first team contract; (b) regular exercise training of > 1 h per day, > 5 days per week for the prior six months; and c) no had to be admitted to hospital due to COVID-19 infection. The following exclusion criteria were applied to potential participants: (a) several injuries within the prior two months; (b) injuries incapacitating football training or matches in the previous 6 months; (c) mechanical instability of the knee joint [13]." Lines 95-101
Reference to the Hägglund study has been added to justify part of these criteria according to the UEFA model, followed in this research (PMID: 15911603).
- Why not study the causes of sports injuries in professional football players?
We welcome the reviewer's expert suggestion to provide the suggested information.
The authors have added subsection 3.2 Injury circumstances in results to report the causes of injury, as follows "One of six-teen muscle and tendons injuries were traumatic with an acute onset. Overuse injuries with a gradual onset were more common among knee injuries than among injuries to the lower back, foot, and hand muscles (50% vs 16.7%, 16.7%, and 16.7%, respectively, p < 0.001). There was only one nerve injury due to L4-L5 disc protrusion. Almost all muscle injuries occurred in noncontact situations (adductors18.2%; quadriceps and hamstrings, 72.7%; and calf muscles, 9.1%), and only 37.5% occurred during matches" Lines 196-203.
- The title should state that it is a study of male professional football players.
The gender male has been added to the title for better understanding as suggested by the reviewer.
- How was the sample size determined? Is the subject's representative of the target population?
Thank you for the suggestion on sample size determination. Based on a recent study by Waldén (PMID: 35552918), the effect size was estimated to calculate the required sample based on the characteristics that were intended for this pilot study, as indicated in the following sentences: “An a priori sample size calculation indicated that participants of a one professional foot-ball team were needed to obtain statistically significant differences in injury incidence rate suffering COVID-19 respect to non-COVID-19 in all session. This a priori sample size was calculated to obtain an effect size of .0.92 in Cohen’s d units (statistical power of 80% with type I error set at 5%), based on a previous investigation that obtained this effect when they study the effect of incidence rate with COVID-19 affected professional football players across 12 countries. [14]. The required sample size was determined using G*Power software [15]”. Lines 101-108.
- The format of Tables 1 through 5 is not three-line, and the horizontal and vertical headings need to be modified.
We appreciate the reviewer's suggestion. Tables 1 to 5 have been modified by three-lines to conform to IEJRPH journal author guidelines.
- The conclusions indicate that the incidence of injury among professional football players infected with COVID-19 is significantly increased compared to the rate of injury in these players before infection, but there are too many confounding factors, and the evidence is not convincing.
Thank you for the reviewer's suggestion. Based on the conclusions and the objective of the study, the paragraph indicating the return to play has been changed to the discussion section and injuries characteristics (Lines 345-353) because it offered confusing data to the conclusions, as the reviewer comments.
However, all the limitations presented in this research are exposed to comment that being the first year of COVID-19 injury study, it was only carried out in a professional football team and with the knowledge demonstrated, these same incidence analyses are being developed in various professional male and female football teams to eliminate these confounding factors that could be included in this research.
On behalf of all co-authors, many thanks for this useful review.

Reviewer 3 Report
Thank you for the opportunity to review this manuscript about injury incidence increases after COVID-19 infection. The study is interesting, although it has some methodological limitations and gaps in the presentation of the collected data. Pre-post COVID-19 period varies slightly from player to player and my major concern is the presentation of the "Match exposure" because of the high match injuries incidence. It would be interesting to present data from different seasons if the authors have long-term cooperation with this club. Please find below some comments for the authors.
Remove double spacing through the manuscript (e.g. lines 72, 137, 159, 258)
Correct the phrase "in a men's professional football players" throughout the manuscript (e.g. line 16, 88)
Table 1. states controls had smaller total exposure than the COVID-19 group. How is that possible if they were not absent for 22 days (median)?
Figure 1. players 8 and 9 have no color difference, please adjust.
Line 125-126 Please explain how was the medical staff previously instructed on how to correctly complete the questionnaire and report all injuries during the season
Line 154 Please erase the 0 after 5 (0.05)
Line 180-181 Was the part “not showing differences about inactivity and return-to-play (p = 0.632)” included in the tables?
The sample consisted of “Twenty-six professional football players, from a team competing in the LaLiga Smartbank”. How often did they play throughout the season? (considering the sample is from one football club - line 299) Please elaborate “match exposure” a little better, not all 26 players have the same match exposure. And of course, please include this in the analysis.
Is there any additional information regarding in-home training programs?
Were there any differences between players with and without symptoms?
In a study by Noya Salces et al. (2014) the authors stated that the incidence of training injuries was greater during the pre-season and tended to decrease throughout the season, while the incidence of competition injuries increased throughout the season – please explain how the results of this study differ from yours because you also found higher match injury incidence later during the season (for example, additionally consider the player who had COVID-19 in September. Did he have a single pre-covid match?)
After considering all the aforementioned please adjust the discussion accordingly.
Author Response
Response to Reviewer comments
ijerph-1857340
Injury incidence increases after COVID-19 infection: a case study with a professional football team
We sincerely thank the Reviewer for carefully proofreading the manuscript and for their helpful and constructive comments. We have addressed all the points raised by the reviewer and have highlighted any change to the manuscript in red using the track changes tool. We believe that our manuscript has been improved by the suggested changes.
Thank you for the opportunity to review this manuscript about injury incidence increases after COVID-19 infection. The study is interesting, although it has some methodological limitations and gaps in the presentation of the collected data. Pre-post COVID-19 period varies slightly from player to player and my major concern is the presentation of the "Match exposure" because of the high match injuries incidence. It would be interesting to present data from different seasons if the authors have long-term cooperation with this club. Please find below some comments for the authors.
Thank you for your positive feedback. Indeed, the collaboration with this club continues and injury incidence data has been collected with two seasons with COVID-19, being in the 2021/2022 season the one in which the omicron variant wave has resulted and we are proceeding with the analysis to present this data in two seasons with delta and omicron variant and extend the evidence presented in this pilot study.
Remove double spacing through the manuscript (e.g. lines 72, 137, 159, 258)
Thank you for the suggestion. The authors have removed the double spaces from the manuscript.
Correct the phrase "in a men's professional football players" throughout the manuscript (e.g. line 16, 88)
Thanks for the suggestion. The phrase " in a men's professional football players " has been changed throughout the manuscript to " in a male professional football players".
Table 1. states controls had smaller total exposure than the COVID-19 group. How is that possible if they were not absent for 22 days (median)?
Controls had less total exposure than subjects with COVID-19 because the injuries they suffered along season kept them out longer training than players with COVID-19. However, the controls had more exposure in matches due to the caution of the players who presented COVID-19 by the team's coaching staff to return-to-play.
Figure 1. players 8 and 9 have no color difference, please adjust.
Thank you for your suggestion. The authors have changed the figure to differentiate the COVID-19 days between player 8 and 9.
Line 125-126 Please explain how was the medical staff previously instructed on how to correctly complete the questionnaire and report all injuries during the season
Thank you for your suggestion. To avoid confusion in the explanation of the collection of injuries by medical personnel, the authors have changed this sentence to " The medical staff of the club recorded injuries during the season ". Lines 142-143.
Line 154 Please erase the 0 after 5 (0.05)
Thanks to the reviewer's suggestion, the 0 after 5 (0.05) has been deleted. Line 172.
Line 180-181 Was the part “not showing differences about inactivity and return-to-play (p = 0.632)” included in the tables?
These data do not include in the tables the periods of inactivity and return to play between players who had COVID-19 and those who did not, because they did not present significant results in the analysis.
The sample consisted of “Twenty-six professional football players, from a team competing in the LaLiga Smartbank”. How often did they play throughout the season? (considering the sample is from one football club - line 299) Please elaborate “match exposure” a little better, not all 26 players have the same match exposure. And of course, please include this in the analysis.
Thank you for your expert suggestion. An analysis of player exposure has been prepared. Indeed, not all players had the same exposure in matches, with players with COVID-19 being less match exposed than players without COVID-19, however, they trained longer than the controls. The match exposure in table 1 has been corrected to avoid the confusion presented by the authors.
However, the injury rate presented in table 3 for match exposure has been checked to confirm the data presented and is correct, so the analysis of the match injury rate is correct, showing that players with COVID-19 did indeed have a higher rate of injury than players without COVID-19, all of them occurred after COVID-19, explained at lines 205-217.
Is there any additional information regarding in-home training programs?
No information about in-home training program has been recorded by the club's physical trainer
Were there any differences between players with and without symptoms?
No differences in injury rate were shown between players with and without COVID-19 symptoms (data not shown).
In a study by Noya Salces et al. (2014) the authors stated that the incidence of training injuries was greater during the pre-season and tended to decrease throughout the season, while the incidence of competition injuries increased throughout the season – please explain how the results of this study differ from yours because you also found higher match injury incidence later during the season (for example, additionally consider the player who had COVID-19 in September. Did he have a single pre-covid match?)
Thank you for your expert suggestion. In this case, the Noya Salces study was carried out in 2008, since that season, the training methods have been modified due to the scientific evidence provided and in turn are data that do not present the casuistry of a disease such as COVID-19, so the results presented by this study are not in accordance with those presented. For this reason, the reference to the study by Noya Salces has been eliminated to avoid this confusion in the discussion and to focus solely on the objective of the study and the results found, marking the incidence in the matches as a differential between the subjects with COVID-19 and those who did not suffer from the disease.
The only player who was infected in September did not play the match prior to illness, suffering the injury in December, being one of the subjects who presented post-COVID-19 injury.
After considering all the aforementioned please adjust the discussion accordingly.
Thank you very much for the accurate revision to improve the submitted manuscript.
On behalf of all co-authors, many thanks for this useful review.
